# Effect of Post–Polyketide Synthase Modification Groups on Property and Activity of Polyene Macrolides

**DOI:** 10.3390/antibiotics12010119

**Published:** 2023-01-08

**Authors:** Liqin Qiao, Yao Dong, Hongli Zhou, Hao Cui

**Affiliations:** 1School of Chemistry and Pharmaceutical Engineering, Jilin Institute of Chemical Technology, Jilin 132022, China; 2College of Biology & Food Engineering, Jilin Institute of Chemical Technology, Jilin 132022, China; 3Engineering Research Center for Agricultural Resources and Comprehensive Utilization of Jilin Province, Jilin Institute of Chemical Technology, Jilin 132022, China

**Keywords:** polyene macrolides, post-PKS modifications, property, antifungal activity

## Abstract

The biosynthesis of polyene macrolides, which are natural products produced by soil actinomycetes, have been extensively explored, and recent studies have focused on the effects of post–polyketide synthase (PKS) modifications to polyene macrolides on toxicity, water solubility, and antifungal activity. For example, there are interactions between glycosyl, carboxyl, and hydroxyl or epoxy groups generated in the post-PKS modification steps; salt bridges will be formed between carboxylate and ammonium on the mycosamine; and water bridges will be formed between hydroxy and hydroxyl on mycosamine. These interactions will affect their water solubility and substrate-recognition specificity. This review summarizes research related to these post-PKS modification groups and discusses some genetic engineering operation problems and solutions that may be encountered when modifying these post-PKS modification groups. In addition, this review provides a basis for the structural research of polyene macrolide antibiotics and contributes to comprehensive and systematic knowledge, and it may thus encourage researchers to develop novel antifungal drugs with higher therapeutic indexes and medical values.

## 1. Introduction

Polyene macrolides are usually composed of macrolactone rings containing 20–40 atoms and mycosamine [1]. Their widely recognized antifungal mode of action occurs by interacting with fungal membrane ergosterol and forming ion channels; however, this mode of action leads to the leakage of cell components, metabolic damage, and, ultimately, cell death [2,3]. This unique antifungal process makes them a class of effective antifungal drugs, which are widely used in the pharmaceutical industry, agriculture, and food preservation. Moreover, this mode of action involves the formation of a prestabilized polyene structure in which the latter is enabled by interactions between the introduced post-PKS modifications and the original macrolactone rings. This in turn enables the antifungals to bind to ergosterol in specific [4,5,6]. The important effects of conjugated double bonds on this mode of action have been elucidated.

Although polyene macrolides have strong antifungal activity and even exert inhibitory effects on some parasites, envelope viruses, and prions, they have serious side effects when frequently used because the low water solubility of polyene macrolides leads to poor distribution in tissues and induces hemolytic activity after binding to cholesterol [3]. In general, toxicity can be reduced and antifungal activity can be improved through structural modification. Therefore, studying the post-PKS modification groups of polyene macrolides through genetic engineering is of great importance to alleviating side effects and enhancing the treatment indexes of these antibiotics.

At present, extensively studied polyene macrolides include mainly nystatin [4,7,8], amphotericin B [9], NPP A_1_ [10], tetramycin [11], 67-121C, pimaricin (also known as natamycin) [12,13], candicidin (also known as FR-008) [14,15], rimocidin [16], filipin [17], aureofuscin [18], S44HP, and nystatin P_1_ (Figure 1a). These antibiotics have similar biosynthesis pathways. PKS repeatedly condenses acetate and propionate units and thereby facilitates the synthesis of carbon chains in a manner similar to fatty acid condensation [19]. Then, cyclization is completed through the catalysis of thioesterase (TE). For example, NysB, NysC, NysI, NysJ, and NysK promote the extension of a polyketone chain of nystatin by condensing 3-methylmalonyl-coenzyme A (CoA) and 15-malonyl-CoA. Finally, the polyketone chain is separated from the acyl carrier protein domain by TE in NysK and cyclized into a nystatin lactone skeleton [20]. The predicted functions of these genes have been confirmed by gene-modification experiments, and a series of antibiotic derivatives has been formulated in this way. These reports have laid a solid foundation for the in-depth study of polyene macrolides. Derivatives obtained through genetic or chemical modification have higher specificity for ergosterol and less for cholesterol, which can reduce the toxicity of the derivatives and preserve or considerably improve their antifungal activities. Significant improvement in the water solubility of derivatives greatly optimizes the pharmacokinetics of these antibiotics and ensures that they are evenly distributed in tissues and easily metabolized by animals. Therefore, polyene macrolides have become attractive models and have been used to propose feasible schemes for new antifungal agents. Here, we will systematically introduce the research progress, structure–activity relationship, and feasible transformation schemes of the post-PKS modification structures of polyene macrolides. Polyene macrolides, polyketide synthase, post-PKS modifications, property, and antifungal activity, therefore, are keywords for this review and searching in Google Scholar.

## 2. Modification of the Carboxyl Group in the Side Chain of Lactone

### 2.1. Effect of Side Chain Carboxyl on Antifungal Activity and Toxicity

As shown in Figure 2, the polyene macrolide pathway has two phylogenetic cytochrome P450 monooxygenase (CYP 450) groups. One group catalyzes the oxidation of a methyl group on the macrolactone ring and includes AmphN, NysN, PimG, and CanC. The other group catalyzes the hydroxylation, leading to the polyol region of the macrolactone ring or the addition of epoxy groups, as seen with AmphL, NysL, TetrK, and PimD.

In the biosynthesis of common polyene macrolides, the first post-PKS modification step is generally carboxylation by using one of the enzymes in Figure 2a [21]. Initially, research on the carboxyl group of polyene macrolides has often focused on the targeted deletion of carboxylases in producers to determine gene function. The influence of the side chain carboxyl group on the antifungal and hemolytic activities of these antibiotics can be explored. As shown in Figure 1b, decarboxylated polyene macrolides produced by gene blocking include mainly 16-descarboxyl-16-methyl amphotericin B, rimocidin C, CE-108C, decarboxylated candicidin, 16-decarboxy-16-methyl nystatin, BSG005, BSG020 (derivative of BSG013), BSG031 (derivative of BSG017), and 12-decarboxy-12-methyl tetramycin B. The antifungal activity of derivatives without the carboxyl group has not changed (about half of the total number), whereas others show reduction or improvement in activity. Table 1 summarizes the changes in their antifungal and hemolytic activities. The minimal inhibitory concentrations (MICs) of 16-descarboxyl-16-methyl amphotericin B and amphotericin B are 1 and 1.25 μg/mL, respectively [22], and the MIC50 (MIC90) values of 16-decarboxy-16-methyl nystatin and nystatin are 1.3 ± 0.4 μg/mL (1.8 ± 0.5 μg/mL) and 1.2 ± 0.2 μg/mL (2.0 ± 0.3 μg/mL), respectively [23]. These results show that their antifungal activity is equivalent to that of the original antibiotics, which were used in antifungal activity tests using Candida albicans strains. Notably, in the antifungal experiment by Carmody et al. [22], the antifungal activity of 16-descarboxyl-16-methyl amphotericin B was slightly higher than that of amphotericin B; they presumed that this difference was due to using combinations of dehydrated and undehydrated forms. After blocking the first type of CYP450 enzyme gene in Streptomyces diastaticus, Seco et al. [24] found that the products rimocidin C and CE-108C showed considerable antifungal activity compared with rimocidin A and CE-108 produced by wild strains; however, no antifungal data were reported. Different from the above derivatives, BSG005, BSG020, and BSG031 against *C. albicans* showed decreased antifungal activities, which were lower than those of S44HP, BSG013, and BSG017. Their structures are shown in Figure 1; their MIC_50_ (MIC_90_) values were 0.07 ± 0.02 μg/mL (0.12 ± 0.03 μg/mL), 0.15 ± 0.03 μg/mL (0.19 ± 0.03 μg/mL), 0.18 ± 0.06 μg/mL (0.37 ± 0.07 μg/mL), 0.12 ± 0.03 μg/mL (0.20 ± 0.03 μg/mL), 0.25 ± 0.07 μg/mL (0.43 ± 0.07 μg/mL), and 0.47 ± 0.15 μg/mL (0.92 ± 0.03 μg/mL), respectively [23]. The original polyene of these decarboxylated derivatives is generally obtained by genetic engineering and is thus not a natural polyene antibiotic. Multiple structural changes lead to complex changes in molecular interactions among the various post-PKS modifications groups of these derivatives and subsequently alter antifungal activity [23]. Some derivatives, such as decarboxylated candidadin, show decreased antifungal activity against *Saccharomyces cerevisiae* Y029. The MICs of decarboxylated candidadin and candidadin are 0.00312–0.00625 and 0.00039–0.00078 μg/mL, respectively, and the antifungal activities of derivatives decrease by 5–10 times [25]. This result may be because carboxyl groups play a more important role in the structural stability and antifungal effect of candicidin. The antifungal activity of pimaricin derivatives was improved in a previous study [26]; the MIC_50_ (MIC_90_) of 12-decarboxy-12-methyl pimaricin against *C. albicans* ATCC 14053 was 0.51 ± 0.01 μg/mL (0.77 ± 0.02 μg/mL), compared with that of pimaricin (1.09 ± 0.02 μg/mL [1.61 ± 0.04 μg/mL]). Against *S. cerevisiae* S288C, the MIC_50_ (MIC_90_) values of 12-decarboxy-12-methyl tetramycin B and tetramycin B were 1.89 ± 0.11 μg/mL (3.11 ± 0.02 μg/mL) and 2.74 ± 0.19 μg/mL (6.01 ± 0.04 μg/mL), respectively, against *Rhodotorula glutinis* CGMCC2.4238; the MIC_50_ (MIC_90_) values of two compounds were 2.26 ± 0.14 μg/mL (5.51 ± 0.02 μg/mL) and 3.95 ± 0.21 μg/mL (8.38 ± 0.03 μg/mL), respectively. The results indicated that the antifungal activity of 12-decarboxy-12-methyl tetramycin B became 1.5–1.9 times that of tetramycin B [27]. Here, we observed that only the antifungal activities of pimaricin, tetramycin B, and candicidin changed after the removal of the carboxyl groups. As shown in Figure 1a, tetramycin B is a relatively simple and short polyene compared with other polyene macrolides, whereas candicidin has a relatively complex structure and long polyene, which may be the reasons why these polyene macrolides are inconsistent upon decarboxylation. Despite the fact that the structure of rimocidin is similar to that of tetramycin B, no obvious change in antifungal activity was observed, and the reason is unknown.

Although changes in the antifungal activities of these polyene macrolides are inconsistent upon decarboxylation, their toxicity decreases, as shown by reduced hemolytic concentrations (HCs) (Table 1). The minimum hemolytic concentration (MHC) of 16-descarboxyl-16-methyl amphotericin B is 50 μg/mL, whereas the MHC of amphotericin B is 5 μg/mL [22]. The hemolytic concentrations causing the 50% hemolysis (HC_50_) of 16-decarboxy-16-methyl nystatin is 175 μg/mL, which is twice that of nystatin (85 μg/mL). The HC_50_ of BSG020 is 9 μg/mL, which is three times that of the BSG013 HC_50_ value (3 μg/mL). The HC_50_ of BSG031 is 0.5 μg/mL higher than that of BSG017 and shows a rising trend [23]. The HC_50_ of 12-decarboxy-12-methyl pimaricin is 478.4 ± 8.58 μg/mL and 4.5 times that of pimaricin (114.0 ± 1.68 μg/mL) [26]. The HC_50_ of 12-decarboxy-12-methyl tetramycin B is 167 ± 4.2 μg/mL, twice the tetramycin B HC_50_ (98.4 ± 6.97 μg/mL) [27]. The precise data of MHC and HC_50_ of rimocidin C and rimocidin A are not shown, but their hemolytic activity data show that the HC value of the former is 2.5–5 times that of the latter [24]. Similarly, the HC of decarboxylated candicidin is 50 times that of candicidin [25].

The presence of the carboxyl group on the polyene macrolides is closely related to their hemolytic activities and is the main reason for their toxicity to mammals. The removal of the carboxyl groups results in a 50-times decrease in hemolytic activity. Carmody et al. [22] have explored this phenomenon. When the polyene interacts with membrane sterols, the carboxyl group is thought to contribute to an extensive network of hydrogen bonds that involves the mycosamine amino group, a water molecule, and the sterol hydroxyl group. Moreover, the network of bonds is equally strong in polyene-cholesterol and polyene-ergosterol complexes, but the removal of the carboxyl group will weaken this network interaction and induces a specific hydrophobic effect, which enables the antibiotics to selectively bind to ergosterol. Palacios [29] directly pointed out the specific mechanism of this interaction; the carboxylate on C16 forms a salt bridge with the ammonium at the C3′ position on the glycosyl. Figure 3 shows this chemical structure. A water-bridged hydrogen bond forms between the hydroxy on C13 and the C2′ hydroxyl moiety on mycosamine. When antibiotics bind to ergosterol and cholesterol, a salt bridge and a water bridge can provide stability for antibiotic molecules, and both the attachment of mycoamine and the relative position of a polyene motif in antibiotics can affect interactions such as a salt bridge and a water bridge. The C2′-deoxyamphotericin B obtained by Wilcock et al. [30] confirmed this finding; refer to Figure 1b for structures. However, limited synthetic access to this derivative has hindered its further development and the determination of whether this improvement in the therapeutic index is coupled with a decreased capacity to evade resistance. Davis et al. [31] replaced the C16 carboxyl group on amphotericin B with some groups containing urea, and the derivatives showed high binding specificity with ergosterol and a high therapeutic index. Among them, amphotericin B methyl urea (amphotericin B MU) and amphotericin B amino urea (amphotericin B AU) showed their high potential and excellent effects on in vivo experiments in mice. Compared with amphotericin B, amphotericin B MU can reduce the fungal burden by 1.2 log units, and amphotericin B AU can reduce the fungal burden by nearly 3 log units (p ≤ 0.0001). Given that this increase in antifungal activity is unexpected, Davis et al. [31] speculated that the reason for this result is that the water solubility of the derivatives improved by more than 20 times. The key point is that the more striking performance of the amphotericin B derivatives developed is their acute toxicity, and the mice did not die even after being injected with 64 mg/kg amphotericin B MU. At the same concentration, the lethal rate of amphotericin B AU reached only 50%, and 4 mg/kg amphotericin B killed the mice in few seconds [31].

The change in antifungal activity in these antibiotics after the removal of the carboxyl group did not show consistency. We speculated that the reasons are differences in the structure and length among antibiotics. The inconsistent changes may also be related to the different test strains used in the antifungal experiments. However, for most of these antibiotics, the removal of the carboxyl groups did not exert a significant impact on their antifungal activities.

### 2.2. Genetic Engineering Methods for the Decarboxylation of Polyene Macrolides

Many problems have been encountered in inducing decarboxylation through traditional genetic engineering strategies. Here, we summarize these problems and sort out different and feasible genetic engineering strategies to prevent future difficulties in blocking similar genes. Figure 4 summarizes changes in these strategic ideas. Initially, all the attempts to induce the decarboxylation of macrolide through *amph*N and *nys*N in-frame deletion have failed because constructed phages cannot be integrated or always produce additional chromosome deletions (Figure 4a) [32]. We encountered the same problem while constructing decarboxylated nystatin and tetramycin producers by destroying *nys*N and *ttm*G. The sequences of these DNA regions have been speculated to be weak recombinations, or these genes deletion strains may produce intermediates that inhibit antibiotic biosynthesis. Carmody et al. [22] inserted another phage gene fragment between *amph*DII and *amph*DI, upstream of *amph*N, to isolate *amph*N from its promoter and inhibit *amph*N expression, obtaining an AmphN mutant for the first time (Figure 4b). Seco et al. [24] blocked *rim*G for the decarboxylation of rimocidin, through gene insertion; however, *rim*H and *rim*A, downstream of *rim*G, were not transcribed, because of the polarity effect, and thus, the mutant stopped producing rimocidin. Furthermore, *rim*H may share a promoter with *rim*G; finally, the C14 decarboxylated derivative of rimocidin was successfully obtained by the complementation of *rim*A in the mutant. The deletion mutant of the *fsc*P gene can be successfully constructed though in-frame deletion without the problems caused by phage integration failure or additional chromosome deletion. However, this method leads to the poor development of spores and a large decline in antibiotic production. To address this problem, Shi [25] opted to target the replacement of specific *Bss*HII fragments in the *fsc*P gene (Figure 4c). The most reliable strategy to decarboxylate polyene macrolides by *nys*N gene disruption is the site-directed mutagenesis strategy adopted by Brautaset [23]. In this strategy, the conservative cysteine residue (the 346th amino acid, responsible for binding to heme) in NysN is replaced with alanine (replacing the coding DNA fragment corresponding to the alanine residue with the restriction enzyme *Hin*dIII digestion site) (Figure 4d). This procedure not only ensures the inactivation of *nys*N but also ensures the normal expression of nearby genes. In previous studies, the same strategy has been successfully used in the decarboxylation of pimaricin and tetramycin B to mutate the conservative cysteine residue (344th and 340th amino acid) in ScnG and TetrG [26,27].

## 3. Research Progress on Side Chain Glycosyl

### 3.1. Glycosylation Mechanism of Polyene Macrolides Antibiotics

Deoxyaminosaccharides contained in polyene macrolides are derived mostly from dTDP glucose [33]. These saccharides are usually the primary metabolites of their production strains and need specific enzymes to catalyze their transformation into the biosynthetic raw materials of polyene macrolides. Caffrey [34] identified three enzymes: phosphomannose isomerase, phosphomannose mutase, and GDP-mannose pyrophosphorylase, which catalyze the synthesis of GDP-mannose (the raw material of mycosamine in amphotericin). The functions of these enzymes are confirmed by observing expected amphotericin lactone products in the gene-blocking strains of these enzymes. In addition, other enzymes that catalyze these raw materials into glycosyl groups and attach to the macrolactone ring have been discovered through protein sequence comparison and in vitro experiments, including AmphDI, AmphDII, and AmphDIII, in the amphotericin pathway, which are the glycosyltransferase, aminotransferase, and GDP-mannose dehydratase, respectively. They catalyze the whole process of modifying intracellular GDP-mannose to the skeleton of the amphotericin B macrolactone ring [33]. Nedal [35] confirmed that NysDI, NysDII, and NysDIII, the homologs of AmphDI, AmphDII, and AmphDIII, respectively, which are involved in the glycosylation of nystatin, are involved mainly in the synthesis and isomerization of GDP-4-keto-6-deoxy-D-mannose and facilitate the connection of “GDP-mycosamine” to the skeleton of nystatin. The glycosylation process is shown in Figure 5.

Xu et al. [36] confirmed that the *fsc*MI gene encodes a glycosyltransferase involved in the biosynthesis of candicidin by gene blocking. The complementation of the homologous glycosyltransferase gene, *amph*DI, *nys*DI, and *pim*K (Figure 2), can restore the transformation of aglycone into candicidin; they identified the conservative amino acid residues in FscMI through site-directed mutation and demonstrate that amino acid residues in glycosyltransferase interact with donors. Ser346, Ser361, His362, and Cys387 are directly involved in the catalysis of FscMI, and the most conserved amino acid residue sites are Ser361 and Cys387. Carrey [33] found that an *amph*DIII deletion mutant was generated by destroying the *Bgl*II digestion site in *amph*DIII to cause the gene frame-shift mutation; the mutant was unable to produce glycosylated amphotericin B, but it disrupted the function of the C8 oxidase of *amph*L, and produced mainly 8-deoxy-amphotericin lactone (Appendix A). The chain reaction of this product change was most likely due to a reduction in the solubility of 8-deoxy-amphotericin lactone after the glycosyl group had been lost. This effect prevented a substrate from reaching the concentration of a normal catalytic reaction, or *amph*L strictly recognized the glycosylated polyene and was unable to add the substrate normally. Only the hydroxylation modification step on C8 occurred after the glycosylation modification step on the amphotericin B-lactone ring. The product of the mutant proves the sequence of the two modification steps. In addition, when other substrates (GDP-D-mannose, dTDP-D-glucose, and even dTDP-L-rhamnose) were used to replace mycosamine, the amphotericin B derivatives produced by the new mutant would have an extremely low yield, which indicated that the substrate compatibility of *amph*DI was low.

### 3.2. Structure–Activity Relationship between Glycosyl and Polyene Macrolides

The glycosyl linked to polyene macrolides is a key component in the antifungal activity of the polyene macrolide. The widely recognized antifungal mechanism of polyene macrolides is the interaction between antibiotic molecules and membrane ergosterols. For example, the mode of action for amphotericin B is that eight antibiotic molecules form an ion channel to cross the membrane, or two of the octamers form an ion channel in the cell membrane in a relative arrangement, causing the leakage of intracellular ions, which lead to an imbalance in intracellular ions and ultimately cause cell death. Figure 6a shows the model of this mechanism [29,37,38,39].

However, pimaricin directly binds to ergosterol to exert an antifungal effect but does not induce membrane permeability [40]. Inspired by this result, Palacios et al. [29] and Kaitlyn [39] found that amphotericin B exists mainly as a large extracellular aggregate that kills yeast by simply combining and extracting ergosterol. Figure 6b shows the combination of amphotericin B and ergosterol, and Figure 6c shows the model of this mechanism. Amphotericin B may kill human cells by combining with cholesterol in a similar model. This binding can affect many physiological characteristics, including vacuolar fusion, endocytosis, pheromone signals, membrane compartmentalization, and the normal function of membrane proteins. The formation of ion channels is a complementary mode of action that can ensure the efficient killing of target fungi. Interestingly, when Daniel [38] and Kaitlyn [39] analyzed the role of post-PKS modification groups in the antifungal activity, the antibiotic derivatives were obtained by chemical synthesis rather than genetic engineering. The performance of these derivatives in in vitro experiments was different from that of the antibiotic obtained by genetic engineering. For example, amphotericin B methyl ester obtained by inactivating the *amph*N gene showed slightly reduced antifungal activity in vitro [22]. However, amphotericin B methyl ester obtained by chemical synthesis was completely consistent with the antifungal activity of amphotericin B in an in vitro antifungal test (the reason for this difference is unclear; amphotericin B used in the antifungal test may have different molecular conformation, or some unknown effects may be induced by gene modification). Meanwhile, the deglycosylated derivatives of amphotericin B and pimaricin obtained by the two methods have almost no antifungal activity, and the deglycosylated amphotericin cannot trigger the membrane permeability of fungi [29,39]. This finding is consistent with the conclusion that other deglycosylated polyene macrolides have almost lost all their antifungal activity and toxicity in in vitro experiments [15,19,41]. These results indicate that additional glycosyls on the macrolactone ring are crucial to their antifungal activity. Palacios et al. [42] found that in the model where amphotericin B bound to ergosterol, the C2′ hydroxyl on mycosamine always formed a key hydrogen bond with the 3β hydroxyl on ergosterol, as shown in Figure 6b. This group may be one of the main reasons for the combination of these antibiotics with ergosterol, but the experimental results quickly overturned this hypothesis, and the results showed that the C2′ hydroxyl did not participate in the combination. Palacios et al. [29] removed the oxygen atom on the hydroxyl group to obtain C2′-deoxyamphotericin B by chemical synthesis, to determine the structure–activity relationship between the group and amphotericin B. The chemical structure is shown in Figure 1. Then, amphotericin B and C2′-deoxyamphotericin B were titrated through isothermal titration calorimetry. The results showed that C2′-deoxyamphotericin B retained the ability to bind to ergosterol, confirming that C2′ hydroxyl did not participate in their combination. However, when the same titration experiment was repeated with cholesterol, the result was the opposite of the previous one. C2′-deoxyamphotericin B was no longer bound to cholesterol, indicating that C2′ hydroxyl was related to the binding of cholesterol. The results of the antifungal test were consistent with those of the titration test. C2′-deoxyamphotericin B retained its antifungal activity against *S. cerevisiae* and *C. albicans* with the MIC of 0.91 μg/mL, and the hemolytic activity was reduced. The MHC and MTC of red blood cells and renal epithelial cells were greater than 454.05 and 72.65 μg/mL, respectively. The changes in antibiotic properties after structural modifications were characterized by the intermolecular interactions of salt bridges and water bridges mentioned above. Here, we show that although the side chain carboxyl group of polyene macrolides participates in this interaction, mycosamine is an indispensable link in these interactions.

The results of the structural modification of the exocyclic carboxyl and glycosyl ligands of polyene macrolides increase what is known about the structure–activity relationship of polyene macrolides. Polyene macrolides bind to ergosterol or cholesterol directly through hydrophilicity or hydrophobicity. The hydrophilicity network involving exocyclic carboxyl groups and mycosamine is crucial to the binding of antibiotics to ergosterol or cholesterol. The structural modification of the exocyclic carboxyl group or mycosamine can weaken this hydrophilicity network, such that the antibiotics can more specifically bind to ergosterol than cholesterol through other action modes and such that the hemolytic toxicity of polyene derivatives is reduced. However, given the similarity of the chemical structure of ergosterol and that of cholesterol, the possibility of such a difference is not significant. Owing to the external carboxyl group, the ammonium at the C3′ position of mycosamine and hydroxyl group at the C2′ position of mycosamine can stabilize the molecular conformation of antibiotics through a hydrogen bond or salt bridge [30,43]. When polyene macrolides maintain their natural conformation, their binding ability to ergosterol or cholesterol is similar, thus showing antifungal activity and hemolytic toxicity. However, when the external carboxyl group or mycosamine is modified, the hydrogen bond or salt bridge effect is destroyed, and the molecular conformation of polyene derivative changes [44]. This new molecular conformation greatly reduces the binding ability of polyene derivatives to cholesterol, but the binding ability to ergosterol remains basically unchanged, so the selectivity of polyene derivatives to ergosterol is improved and the hemolytic toxicity is reduced.

### 3.3. Research Progress on the Second Glycosyl of Polyene Antibiotics

In addition to modifying the originally linked glycosyl on the macrolactone ring, linking additional glycosyl to the original glycosyl can improve the pharmacological properties of these antibiotics. The disaccharide polyene macrolide NPP A_1_ was first found in *Pseudomonas autotrophica*. Given that all the polyene macrolide biosynthesis gene clusters currently characterized encode the cytochrome P450 (CYP) hydroxylase gene, Lee [10] amplified the highly conserved region of about 350 bp in this gene through polymerase chain reaction on *P. autotrophica* to determine whether the strain’s chromosome encodes the polyene biosynthesis gene. In addition, a detailed sequence analysis of the gene cluster found that it was highly homologous with the nystatin biosynthesis gene cluster in *S. noursei* [7]. The difference between the two clusters is that *npp*M, speculated to correspond to the *nys*M gene, is at the regulatory region on the right side of the *npp* biosynthesis gene cluster rather downstream of the *npp*N gene. Moreover, additional gene *npp*O, which may encode acyl CoA decarboxylase, was found in this region. Given that NPP A_1_ shows earlier elution time than nystatin in a PDA-HPLC analysis, the NPP A_1_ aglycone part is the same as nystatin but contains a different modification from nystatin and different glycosylation modes [45]. The 1D and 2D nuclear magnetic resonance (NMR) and MS results showed that the structure of NPP A_1_ consists of a macrolide aglycone part identical to nystatin and a unique disaccharide part, mycosamine (α 1-4)-N-acetylglucosamine, as shown in Figure 1a. To determine the influence of disaccharide on the properties of antibiotics, the solubility of NPP A_1_, nystatin as control, and in a 10 mmol Tris-HCl solution (pH 7.0) was tested through spectrophotometry [46]. The solubility of NPP A_1_ and nystatin were 34.7 mg/mL and 0.11 mg/mL, respectively. Therefore, the water solubility of NPP A_1_ containing an additional N-acetylglucosamine moiety was more than 300 times that of nystatin. The MIC_50_ values of NPP A_1_ and nystatin against *C. albicans* ATCC10231 were 1.08 μg/mL and 0.43 μg/mL, respectively, and their HC_50_ values were 403.7 μg/mL and 33 μg/mL, respectively. These results indicate that the antifungal activity of NPP A_1_ is about half that of nystatin, and the hemolytic toxicity is 1/10 that of nystatin. Polyene macrolides carrying disaccharides have few side effects and can thus be developed into new drugs [46]. Kim [47] found that the genome of *P. autotrophica* has 112 putative glycosyltransferase genes. An analysis of the position of these genes in the NPP A_1_ biosynthetic gene cluster has shown that one of the glycosyltransferase genes is similar to the previously reported second glycosyltransferase NypY, so a second glycosyltransferase in NPP A_1_ biosynthesis is presumed and named *npp*Y. The gene is located at the edge of the NPP A_1_ biosynthesis gene cluster. In addition, this gene has 51% identity with the second glycosyltransferase PegA responsible for 67-121C (Figure 1a) in *Actinoplanes caeruleus*. The glycosyltransferase NppY responsible for the addition of the second glycosyl group in the NPP pathway has been identified [47]. The appearance of 10-deoxy nystatin lactone and the expected disaccharide products in the targeted deletion and complementation of the *npp*Y gene indicates that NppY has the same function as glycosyltransferase. The lack of a C10 hydroxyl group indicates that *npp*L recognizes disaccharide antibiotic precursors in specific. Appendix A shows changes in NPP A_1_ molecules after the blocking of *npp*L [47]. The site-directed mutagenesis indicates that the amino acid residue site (R200N) is crucial to the functional structure of NppY. The sequence alignment results of *peg*A and *npp*Y obtained by Stephens [48] are consistent with those of Kim [47]. Barke [49] found a nystatin-like disaccharide polyene macrolide produced by *Pseudonocardia* sp. P1, isolated from *Acromyrmex octospinosus* garden worker ants. The compound is different from NPP and named nystatin P1. The sequencing of its genome shows that the PKS genes contained in the nystatin P1 pathway has more than 90% identity with the PKS genes in NPP synthesis. However, a second glycosyltransferase (NypY) in the nystatin P1 pathway is not present in *S. noursei* and *P. autotrophica*. NypY belongs to the same family as NypDI but has only 41% identity. Nystatin P1 was analyzed by HPLC-MS and compared with the nystatin standard. They had the same aglycone part, but the difference in molecular weight between them was 162, indicating that nystatin P1 had an additional hexose group. Secondary mass spectrometry showed that a disaccharide component is attached to the macrolactone ring rather than to two glycosyls at different positions. To further study the action mechanism of the second glycosyltransferase and the substrate selectivity, the second glycosyltransferase gene *nyp*Y was heterologously expressed in *S. nodosus* and its mutants. NypY transformed amphotericin A, amphotericin B, and 7-oxo amphotericin B into disaccharide-modified forms (Figure 7) in the 5% transformation amount of the yield of polyene [50]. However, when only amphotericin derivatives containing no extra carboxyl group or no amino group on mycosamine existed in a cell, no additional disaccharide antibiotics were produced, indicating that NypY eliminated the two groups. Only the precursor completely modified by these two groups can be used as a substrate of NypY for the addition of a second glycosyl. In addition, the heterologous expression of *nyp*Y and *pegA* in *S. albidoflavus* DSM40624 indicates that NypY and PegA can recognize and add a second sugar group to candicidin [50]. The derivatives’ structures are shown in Figure 7. Walmsley [28] inserted a second glycosyltransferase synthesis gene *nyp*Y into *S. nodosus* Δ*amph*L, the producer of 8-deoxyamphenicol B (a previous study showed that 7-oxo amphotericin B is a more suitable substrate for NypY than amphotericin A and B are) and found that mannosyl-8-deoxyamphotericin B (Figure 7) accounted for about 20% of the total heptaene compounds produced. Mannosyl amphotericin A accounts for about 40% of the total tetraene (*S. nodosus* produced two polyenes, tetraene amphotericin A, and heptaene amphotericin B). This result shows that amphotericin A is more easily recognized than tetraenoid amphotericin A by NypY, and mannosyl-8-deoxyamphotericin B is more easily recognized than amphotericin B by NypY (amphotericin B has only 5% conversion rate). Therefore, 8-deoxy amphotericin B is a more suitable substrate for NypY than amphotericin B is. The antifungal activity and hemolytic toxicity of mannosyl-8-deoxyamphotericin B have been tested. Its antifungal activity is the same as that of amphotericin B, with MIC_50_ values of 1.46 μg/mL, but the hemolytic activity of amphotericin B is 1.6–2.5 times that of mannosyl-8-deoxyamphotericin B. The MHC_0_, MHC_50_, and MHC_100_ of amphotericin B are 0.58 μg/mL, 1.46 μg/mL, and 2.92 μg/mL, respectively, and those values of mannosyl-8-deoxyamphotericin B are 0.94 μg/mL, 2.35 μg/mL, and 7.43 μg/mL, respectively. These data indicate that the addition of a second glycosyl group indeed reduces the toxicity and side effects of 8-deoxyamphtericin B.

The roles of the second glycosyl are mainly increasing water solubility and reducing the hemolytic activity of antibiotics. These changes usually come at the expense of partial antifungal activity. However, owing to a considerable decrease in the side effects of antibiotics, these findings are promising. Research on structural modification related to the second glycosyl group has focused mainly on the second glycosyltransferases, such as NppY, NypY, PegA, and OleD. Researchers have attempted to convert these glycosyltransferases into a variety of antibiotic producers to obtain disaccharide polyene macrolide derivatives. These enzymes have a certain substrate breadth and can recognize a variety of polyene macrolides, but their conversion rates are limited. Nevertheless, they have high recognition efficiency for polyene macrolides removed by specific hydroxyl groups (8-deoxy amphotericin B) [28]. These second glycosyltransferases are highly homologous to the first glycosyltransferases, such as FscMI, TtmK, and NysDI (Figure 2c), and have high recognition rates for dehydroxylated nystatin and amphotericin B, similar to the first glycosyltransferases. Therefore, the action mechanism of the second glycosyltransferases is similar to that of the first glycosyltransferases. The second glycosyltransferases often precede hydroxylation or epoxidation modification and have certain recognition specificity for antibiotic precursors without involving modified groups. The conversion efficiency of these glycosyltransferases may be increased by blocking the hydroxylation or epoxidation of the CYP450 enzyme, integrating a second glycosyltransferase, or inducing hydroxylase or epoxidase in a blocking strain. Improving the conversion rate of the second glycosyl may be a research topic of interest in the future.

## 4. Research Progress on Epoxidation or Hydroxylation of Macrolactone Ring

Parts of the hydrophilic groups in the polyol region of polyene macrolides are completed through the hydroxylation or epoxidation of post-PKS modifications. Enzymes involved in these modes of catalysis are mainly from the second type of P450 monooxygenases (AmphL, NysL, ScnD, and PimD) encoded by the genes in its biosynthetic gene cluster [51]. Their evolutionary histories are shown in Figure 2. These groups can often improve the stability of the interactions between antibiotics and membrane ergosterols. Therefore, a series of structurally modified derivatives of the polyol region of polyene macrolides have been obtained by modifying P450 enzyme genes.

Aparicio [52] revealed that *pim*D is located downstream of the biosynthetic gene cluster of pimaricin and has high identity with other CYP450 monooxygenase genes. The *pimD* deletion strain *S. natalensis* 6D4 is generated by mediating with phage; 4,5-deoxypimaricin (Figure 8) is its only product, according to HPLC and LC-MS analyses. The 4,5-deoxypimaricin to some extent retained antifungal activity, and its MIC is 70 μg/mL, which is 10 times that of pimaricin [53]. In the amphotericin B pathway, AmphL is homologous to NysL (responsible for introducing the C10 hydroxyl group of nystatin [51]) and to PimD (responsible for introducing epoxy to the polymer chain of pimaricin). Byrne [33] disrupted *amph*L in *S. nodosus*; the heptaene and tetraene yields of the mutant were greatly reduced (2–5 mg/L) compared with the amphotericin yield of by *S. nodosus* ATCC14899. After electrospray ionization mass spectrometry and an NMR analysis, the heptaene product was determined to be 8-deoxyamphotericins (Figure 8). The antifungal activity of the compound was one-fourth of the purified amphotericin B, with MICs of 1.25 μg/mL and 0.3125 μg/mL. In addition, Carmody et al. [22] demonstrated that the hemolytic activity of 8-deoxyamphotericins is lower than that of amphotericin B. The 10-deoxynystatin (Figure 8) was produced by the *nys*L destruction strain, and it showed the same antifungal activity as nystatin; the MIC_50_ values for *C. albicans* were all 0.45 μg/mL [54]. This result was the result of a structural change that may have affected the effect of new compounds on the antifungal activity of *C. albicans*. Notably, their experiment involved the positive correlation between the amount of NysL and the nystatin synthesis process. If NysL accumulation in a cell is insufficient, nystatin production is reduced. As shown in Figure 1b, the DH15 domain in NysJ was destroyed, the C9–C10 region of nystatin changed, and NysL was unable to correctly identify the nystatin precursor, failed to catalyze the hydroxylation of nystatin precursor C9, and produced a derivative named BSG002. The antifungal activity of BSG002 relative to nystatin was reduced by four times, and the hemolytic activity was reduced by two times. The MIC_50_, MIC_90_, and HC_50_ values of BSG002 (nystatin) were 4.8 ± 0.4 (1.2 ± 0.2) μg/mL, 2.0 ± 0.3 (10.5 ± 3.5) μg/mL, and 180 (85) μg/mL, respectively. The results are consistent with the inference that the hydrophilicity of a channel may be affected by the number and location of hydroxyl groups. Brautast [23] mentioned that the addition of the hydroxyl group at C9 will increase conductivity and thus increase antifungal activity, whereas the removal of the hydroxyl group at C10 will have little effect on antifungal activity. The decline in antifungal activity does not seem to support this view.

Santos [55] explored the substrate-recognition specificity of CYP450 enzymes, such as NysL, and integrated homologous genes, such as *amph*L, *nys*L, and *tetr*K (transforming tetramycin A into tetramycin B), into *S. natalensis* 6D4 to explore whether their encoding products can catalyze the epoxidation or hydroxylation of 4,5-deoxypimaricin. Except for AmphL, CYP450 enzymes can recognize 4,5-deoxypimaricin and produce 6-hydroxy-desepoxypimaricin (6-OH-DEP; see Figure 8) and pimaricin. However, their catalytic efficiency is low. NysL can convert 23% (156 ± 21 μg/mL) of 4,5-deoxypimaricin into 6-OH-DEP, whereas TetrK can convert only 18% (89 ± 9 μg/mL) of 4,5-deoxypimaricin into pimaricin. This result indicates an imperfect active site recognition. The *nys*L was placed under the control of a double promoter combining the *pim*D and *erm*E* promoters into pIB139. This method can improve the catalytic efficiency of NysL to 48% (341 ± 64 μg/mL); they deduced the reasons why the catalytic efficiency of the two CYP450 enzymes are lower than PimD. First, the catalytic efficiency of the two enzymes is different. Second, for TetrK, the efficiency of catalytic epoxidation is extremely low. As for NysL, the differential substrate size is not conducive to NysL recognition, and NysL may interact positively with a substrate. The antifungal activity results show that the MIC_50_ values of the 6-OH-DEP and 4,5-deoxypimaricin are 47.1 μg/mL, which is 15 times that of pimaricin. In addition to the experiment on the modification of *nys*L of Volokhan [56], the antifungal activity and hemolytic activity of the de-epoxidation or decarboxylation-modified derivatives of other reported polyene macrolides are lower than those of the corresponding starting antibiotic molecules. That is, the chemical structures of the polyol region of these antibiotics are crucial to their ability to interact with membrane ergosterol or cholesterol. According to the polyene ion channel model, the polyol region of polyene macrolides is in the ion channel. The change of hydrophilic groups in the polyol region not only affects the characteristics of the ion channel but also affects the hydrogen bond network within an antibiotic molecule and the conformation of the antibiotic molecule, thus affecting the pharmacological properties of polyene macrolides. In addition, the second type of CYP450 monooxygenase has relatively high catalytic specificity. For example, AmphL cannot recognize amphotericin B in the absence of glycosyl modification. By contrast, NysL and TetrK seem to be much less specific than AmphL. Although NysL cannot identify the correct position after changes in the C9–C10 region of the nystatin precursor, it can still catalyze. NysL and TetrK can catalyze a small amount of other polyene macrolide precursors. These results reflect that the substrate-recognition specificity of some homologous CYP450 monooxygenases is not as high as expected.

## 5. Conclusions

In studies on the post-PKS modification of polyene macrolides, the absence of the post-PKS modification carboxyl group or the presence of additional glycosyl decreases the toxicity of natural antibiotics. However, the effect of removing the post-PKS modification group of antibiotics on antifungal activity is uncertain, because they are closely related to the conformation stability, hydrophilicity, and conductivity of antibiotic molecular structure. They affect the affinity of antibiotic molecules with membrane ergosterol or cholesterol. In addition, the number of conjugated double bonds on the macrolactone ring often affects the rigidity, hydrophobicity, and stability of the ion channel of an antibiotic molecule. These molecular structures are also closely related to the antifungal activity and toxicity of polyene macrolides. Therefore, obtaining positive examples and combining a variety of post-PKS modifications results in the development of novel polyene macrolides with low toxicity and excellent efficacy. Furthermore, summarizing the effect of a post-PKS modification group on the structures and activities of polyene macrolides will help us obtain the desired antibiotic products. It is believed that these drugs have the potential efficiency of solving fungal infection, and the reduction of toxicity has far-reaching significance for drug safety.

## Figures and Tables

**Figure 1 antibiotics-12-00119-f001:**
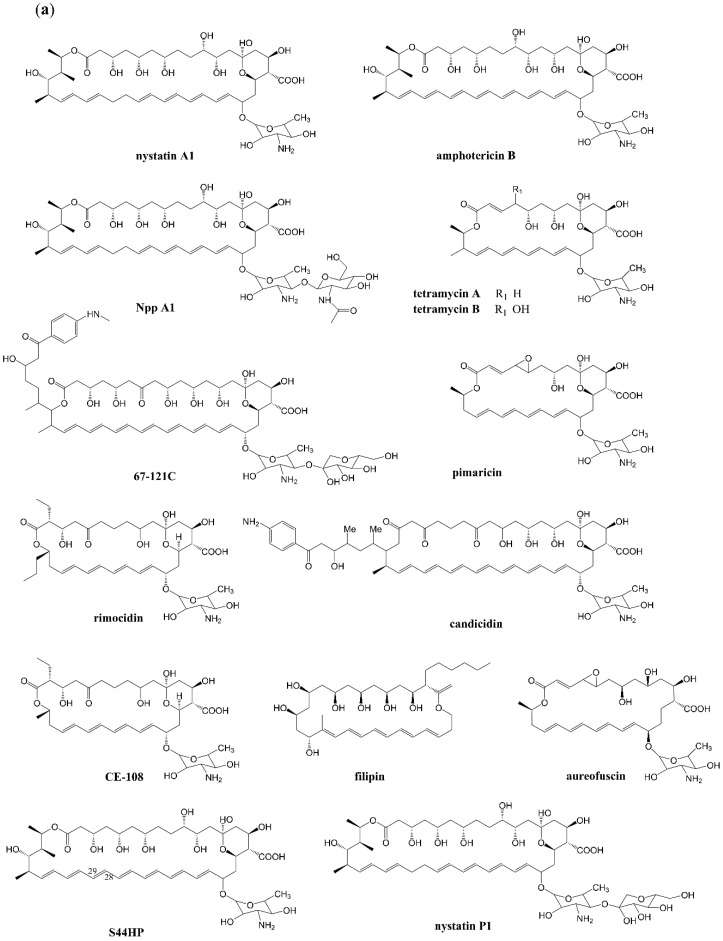
Structures of polyene macrolides and they derivatives. (**a**) Initial polyene for structural modification. (**b**) Polyene related to the modification of the carboxyl group, a derivative C2′-deoxyamphotericin B modified on the glycosyl, is added to facilitate the structure comparison; highlights represent differences from the initial compound.

**Figure 2 antibiotics-12-00119-f002:**
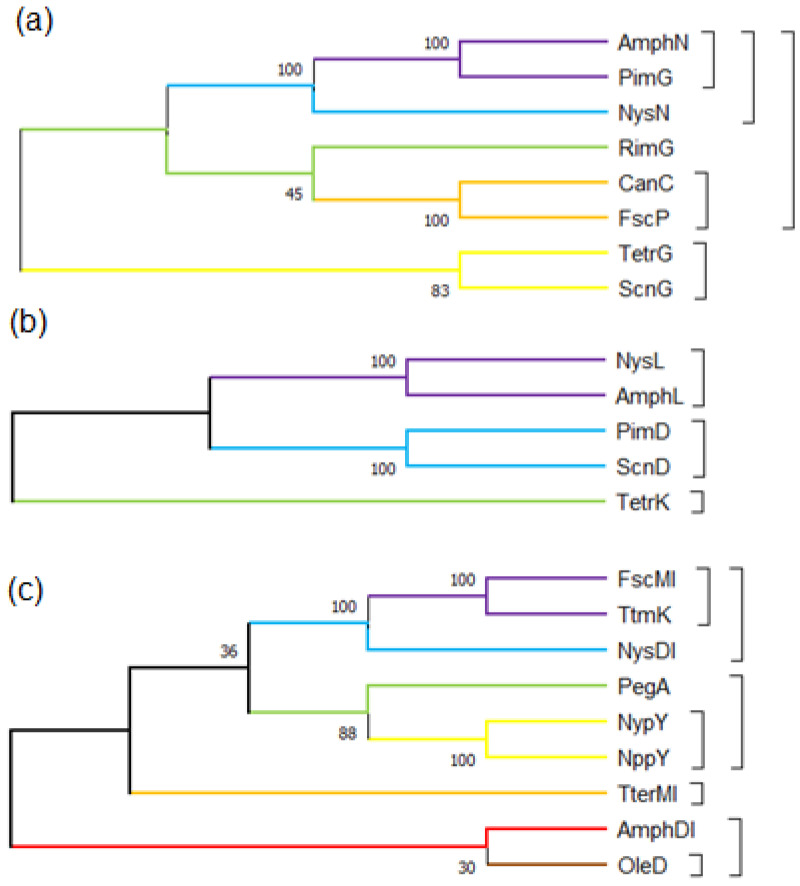
Phylogenetic tree of the post-PKS modifying proteins of some polyene macrolides. The evolutionary history was inferred using the neighbor-joining method. (**a**) Phylogenetic tree of carboxylating CYP450 enzymes; (**b**) phylogenetic tree of hydroxylating or epoxidizing CYP450 enzymes; (**c**) phylogenetic tree of glycosyltransferase.

**Figure 3 antibiotics-12-00119-f003:**
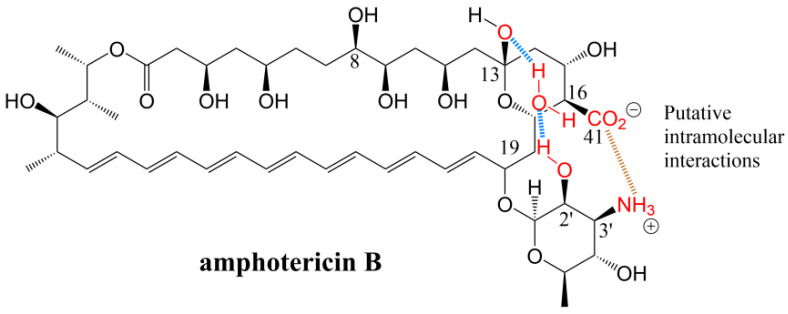
Intramolecular interactions of amphotericin B. The blue part represents the water bridge formed by water molecules between the hydroxy on C13 and the C2′ hydroxyl moiety on mycosamine, and the brown part represents the salt bridge formed by carboxylate on C16 and the C3′ position on the glycosyl [31].

**Figure 4 antibiotics-12-00119-f004:**
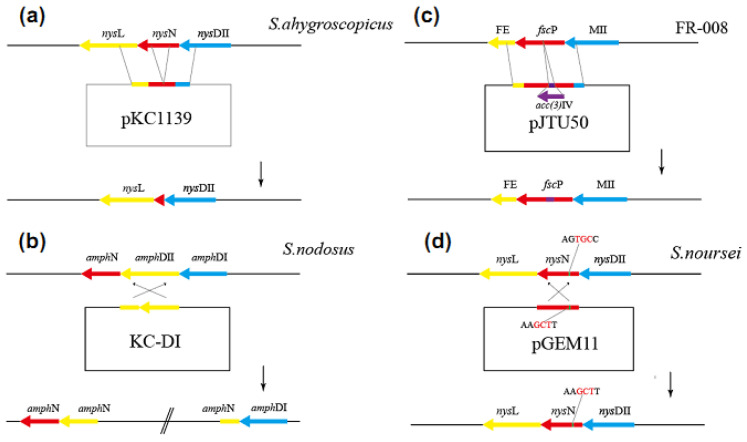
Genetic engineering–based strategies for the decarboxylation of macrolide. (**a**) This is a classic in a frame-deletion scheme. This scheme will lead to phage failure to integrate or additional chromosome deletion, which cannot achieve the effect of deleting only the target gene. Therefore, no specific diagram has been reported in the literature. The following is the thought map of our laboratory. (**b**) Insert additional gene fragments into the upstream of the target gene to separate its promoter from the gene, thus preventing the transcription of the gene, and then supplement the gene destroyed by the additional insertion to achieve the effect of targeted deletion. This scheme for the first time realizes the translation blocking of specific CYP450 enzymes. (**c**) A new gene insertion scheme directly inserts additional apramycin-resistant genes into the conservative sequence of the target gene to achieve the effect of targeted deletion. However, this scheme may have a polar effect on the transcription of downstream genes, leading to the failure of the normal transcription of downstream genes, but the affected downstream genes still need to be supplemented. (**d**) In this scheme, the reliable way to block the translation of specific CYP450 enzyme is to first compare the amino acid active site of this protein and then mutate the codon from this site to an enzyme digestion site. This method can not only ensure the inactivation of the target gene without affecting the upstream and downstream genes but also facilitate the construction of gene-blocking plasmids.

**Figure 5 antibiotics-12-00119-f005:**
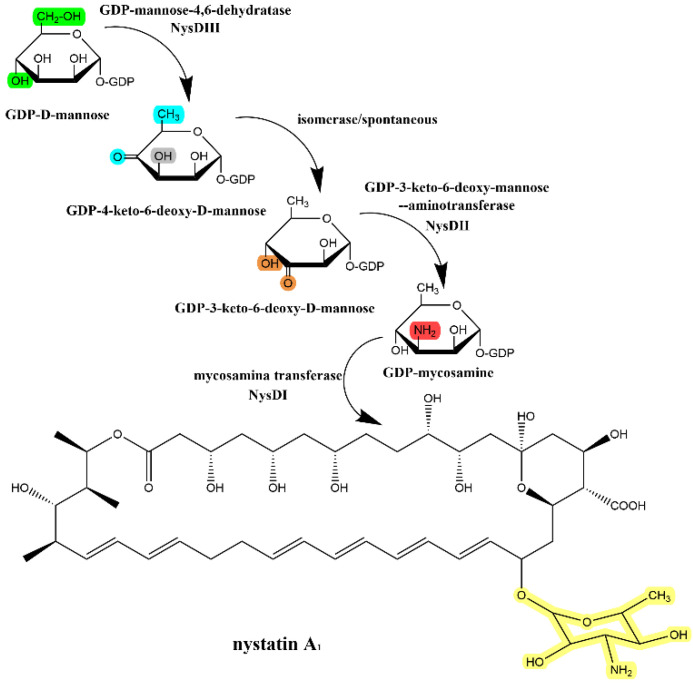
Addition of mycosamine in the biosynthesis of nystatin. The highlights show the differences before and after each catalytic step.

**Figure 6 antibiotics-12-00119-f006:**
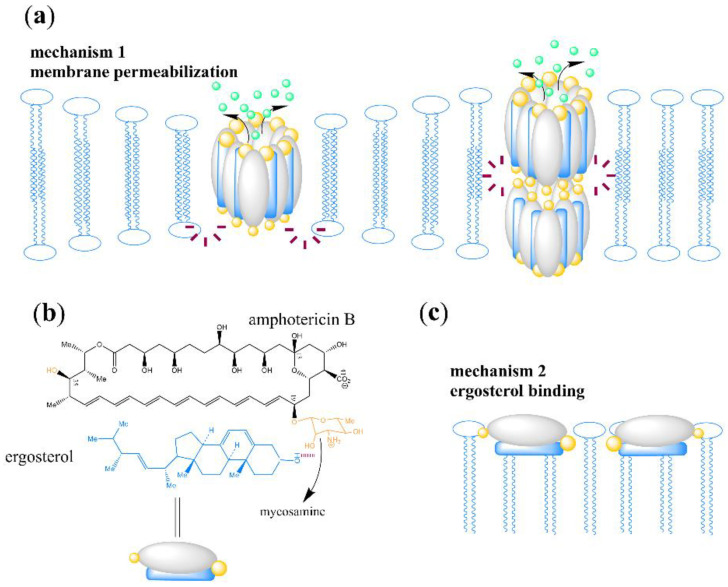
Model and antifungal mechanism of amphotericin B binding to ergosterol. (**a**) The first kind of antifungal mechanism of amphotericin B, membrane permeabilization; (**b**) the model of amphotericin B, ergosterol binding; (**c**) the second of antifungal mechanism of amphotericin B, ergosterol binding [39].

**Figure 7 antibiotics-12-00119-f007:**
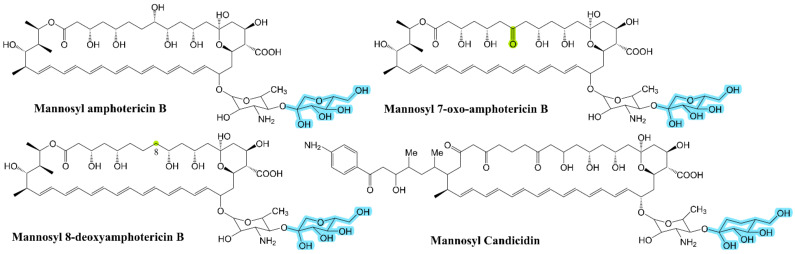
Polyene structure with additional glycosyl added by NypY; highlights represent differences from the initial compound.

**Figure 8 antibiotics-12-00119-f008:**
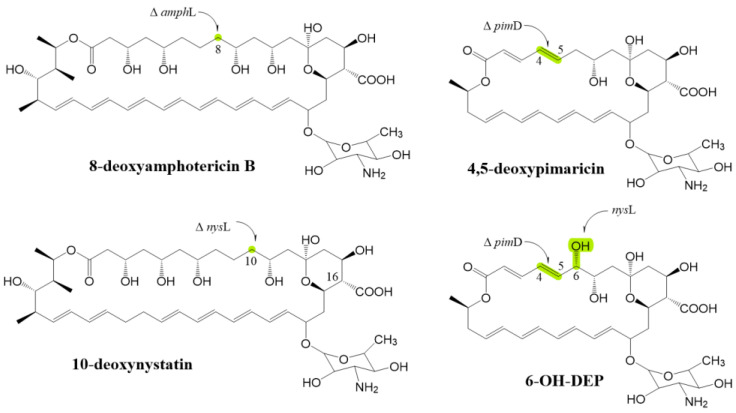
Structure of polyene macrolide derivatives related to hydroxyl; highlights represent differences from the initial compound. 6-OH-DEP is 6-hydroxy-desepoxypimaricin.

**Table 1 antibiotics-12-00119-t001:** The antifungal activity and hemolytic activity of the polyene macrolide and their derivatives [20,21,22,23,24,28].

Compound	Antifungal Activity		Test Strain	Hemolytic Activity	
amphotericin B	MIC	1.25 μg/mL	*Candida. albicans*	MHC	5 μg/mL
16-descarboxyl-16-methyl amphotericin B	MIC	1 μg/mL	*C. albicans*	MHC	50 μg/mL
nystatin	MIC_50_ MIC_90_	1.2 ± 0.2 μg/mL 2.0 ± 0.3 μg/mL	*C. albicans* ATCC 14053	HC_50_	85 μg/mL
16-descarboxyl-16-methyl nystatin	MIC_50_ MIC_90_	1.3 ± 0.4 μg/mL 1.8 ± 0.5 μg/mL	*C. albicans* ATCC 14053	HC_50_	175 μg/mL
S44HP	MIC_50_ MIC_90_	0.12 ± 0.03 μg/mL 0.20 ± 0.03 μg/mL	*C. albicans* ATCC 14053	HC_50_	2.5 μg/mL
BSG005	MIC_50_ MIC_90_	0.07 ± 0.02 μg/mL 0.20 ± 0.03 μg/mL	*C. albicans* ATCC 14053	HC_50_	4.0 μg/mL
BSG013	MIC_50_ MIC_90_	0.25 ± 0.07 μg/mL 0.43 ± 0.07 μg/mL	*C. albicans* ATCC 14053	HC_50_	3.0 μg/mL
BSG020	MIC_50_ MIC_90_	0.15 ± 0.03 μg/mL 0.19 ± 0.03 μg/mL	*C. albicans* ATCC 14053	HC_50_	9.0 μg/mL
BSG017	MIC_50_ MIC_90_	0.47 ± 0.15 μg/mL 0.92 ± 0.03 μg/mL	*C. albicans* ATCC 14053	HC_50_	3.3 μg/mL
BSG031	MIC_50_ MIC_90_	0.18 ± 0.06 μg/mL 0.37 ± 0.07 μg/mL	*C. albicans* ATCC 14053	HC_50_	3.8 μg/mL
rimocidin A			Same antifungal activity, data not shown		
rimocidin C			Same antifungal activity, data not shown	2.5–5 times that of rimocidin	
CE-108			Same antifungal activity, data not shown		
CE-108 C			Same antifungal activity, data not shown		
candicidin	MIC	0.00039–0.00078 μg/mL	*Saccharomyces cerevisiae* Y029		
decarboxylated candicidin	MIC	0.00312–0.00625 μg/mL	*S. cerevisiae* Y029	50 times that of candicidin	
pimaricin	MIC_50_ MIC_90_	0.51 ± 0.01 μg/mL 0.77 ± 0.02 μg/mL	*C. albicans* ATCC 14053	HC_50_	114.0 ± 1.68 μg/mL
12-decarboxy-12-methyl pimaricin	MIC_50_ MIC_90_	1.09 ± 0.02 μg/mL 1.61 ± 0.04 μg/mL	*C. albicans* ATCC 14053	HC_50_	478.4 ± 8.58 μg/mL
tetramycin B	MIC_50_ MIC_90_	2.74 ± 0.19 μg/mL 6.01 ± 0.04 μg/mL	*S. cerevisiae* S288C		
12-decarboxy-12-methyl tetramycin B	MIC_50_ MIC_90_	1.89 ± 0.11 μg/mL 3.11 ± 0.02 μg/mL	*S. cerevisiae* S288C		
tetramycin B	MIC_50_ MIC_90_	3.95 ± 0.21 μg/mL 8.38 ± 0.03 μg/mL	*Rhodotorula glutinis* CGMCC2.4238	HC_50_	98.4 ± 6.97 μg/mL
12-decarboxy-12-methyl tetramycin B	MIC_50_ MIC_90_	2.26 ± 0.14 μg/mL 5.51 ± 0.02 μg/mL	*R. glutinis* CGMCC2.4238	HC_50_	167.0 ± 4.20 μg/mL
amphotericin B	MIC_50_	1.44 μg/mL	*C. albicans ATCC10231*	MHC_0_ MHC_50_ MHC_100_	0.58 μg/mL 1.46 μg/mL 2.92 μg/mL
mannosyl-8-deoxyamphotericin B	MIC_50_	1.46 μg/mL	*C. albicans ATCC10231*	MHC_0_ MHC_50_ MHC_100_	0.94 μg/mL 2.35 μg/mL 7.43 μg/mL

MIC, MHC, and HC represent the minimal inhibitory concentrations, the minimum hemolytic concentrations, and hemolytic concentrations, respectively. MIC_50_ and MIC_90_ represent minimum inhibitory concentrations causing 50% and 90% inhibition of cell growth, respectively. MHC_0_, MHC_50_, and MHC_100_ represent the concentrations giving 0%, 50%, and 100% hemolysis, respectively. HC_50_ represents hemolytic concentrations causing 50% hemolysis.

## Data Availability

Not applicable.

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
