# Peer review of "Effect of Post–Polyketide Synthase Modification Groups on Property and Activity of Polyene Macrolides"

_antibiotics, 2023, doi:10.3390/antibiotics12010119_

Round 1

Reviewer 1 Report

The Manuscript by Cui et al. concerns the reviewing the main “structure – activity” features for some polyene macrolides. There are 56 references, which include literature sources of 1964-2020 years. I think that this or related articles are useful, interesting for a wide readership of Antibiotics journal and specialists working in different fields of science (biology, medicine, chemistry etc.).

At the same time there are several important notes to the Authors, concerning mainly the language (including English language polishing) and several incorrect statements (see below). After fixation of these problems, I advise to Accept this Manuscript in revised form (Minor Revision).

Notes to Authors:

1)      The phrases, which should be changed:

a.       Abstract, line 17: “…molecular (?) interactions among (?) post-modified (in this context, it is not clear which ones) groups”. Intramolecular interactions?

b.      Conclusions, lines 579-580: “…absence of a post-modified or the presence…”

c.       Line 582: “… these conditions” Which ones?

d.      Line 582: “…stability of antibiotic molecular structure” (?) Are the compounds unstable?

e.      Line 584: “… some structures (?) in the synthesis process (???) (there are no structures in synthesis)”.

f.        Line 96: “… some derivatives without carboxyl groups, some have…” (???)

g.       Line 185: “…height potential”.

h.      Line 303: should be “structure – activity”.

2)      I haven’t SI file for this Manuscript.

3)      Figure 3+ line 173: “… acetate (?) on the side chain”. There is no Acetate (!); but carboxylate is really present. Furthermore, salt bridge could only be formed between carboxylate, RCO2(-), and ammonium (protonated amine), R’NH3(+) (not free amine).

4)      Lines 175-175: “…hydrogen bond between the hydrogen energy (sic!)” ??? Moreover, there is typo (mistake?) in structure in Fig. 3 with OH etc. Besides, the presentation of OH-13 and OH-2’ groups is not convenient for bonding (they are trans-).

5)      Figure 6: there are numerous mistakes (“brige”; notations of the interactions are mixed up).

6)      References 1, 2, 10, 18, 43, 53 should be written according to https://cassi.cas.org/search.jsp.

Reviewer 2 Report

The review of Liqin Qiao et al. tries to describe the effects of post-synthesis modifications introduced on polyene macrolides on their physiological properties and biological effects. The authors gathered a lot of information on this interesting topic, and tried to communicate their results with this manuscript.

Unfortunately, at this stage the paper should be rejected as of language problems.

 I advise to completely reformulate and rewrite the text, have a language check by a native speaker, and to subsequently resubmit the paper. At the moment large parts of the text are very difficult to grasp and sometimes even claim the opposite of what authors probably mean to state. I honestly gave up trying to correct after some pages, as this is not our job.

I here below state some remarks (not all!) which I assembled on these first pages only to showcase the difficulties.

First of all, everywhere in the paper including in the title, the term “post-modification groups” or “post-modified groups” is used, while actually is meant “modifications introduced post-synthesis of the polyene structure”. Hence these are “post-synthesis modifications” of polyene structures. Please correct everywhere.

Line 18: there are no molecules “on” the glycosyl groups. There are salt and water bridges formed between molecules “via” glycosyl groups…

Line 25: correct to “encourage researchers to develop”.

Line 32: correct to “Their widely recognized antifungal mode of action is by interacting with fungal membrane ergosterol and forming ion channels”

Line 37-38: “interaction between the post-modified polyketide synthase (PKS) groups and macrolactone rings”?  There are no interactions between PKS groups and the macrolactone rings. I presume the authors mean “This mode of action involves the formation of a pre-stabilized polyene structure in which the latter is enabled by interactions of the introduced post-synthesis modifications with the original macrolactone rings. This in turn enables the antifungals to specifically bind ergosterol …”.

Line 50-53: I would include figure S1 in the manuscript, as the reader otherwise has no clue yet what kind of structures are listed here.

Figure S1 (to be included in manuscript): why is the final sugar in the nystatin P1 structure highlighted in blue? Elaborate in your review. The same holds for the purple double bond at C28-C29 for S44HP, and extensions for rimocidin and CE-108 in brown?

Figure S1: The list at lines 50-53 does not fully correspond with the figure S1?

Figure S1: when including Fig S1, include the abbreviation "AmB" with the amphotericin structure, allowing to understand the structure names as found in figure 2.

Figure S2 likewise could be incorporated to clearly show the structural differences. Likewise elaborate on the colors used to indicate parts of the various structures.

Line 61: eliminate “of the”. (correct to “the predicted functions”)

Line 62: correct to “and this way”.

Line 64-65: correct to “have higher specificity for ergosterol and less for cholesterol”.

Line 65-66: correct to “which can reduce the toxicity of the derivatives that preserved their antifungal activities or even can lead to significantly improved activity.”

Line 68: correct to “of these antibiotics and ensure that they are ….”.

Line 70: correct to “to propose feasible schemes for new antifungal agents…”.

Line 72: correct to “schemes”.

Line 74: correct to “…therefore are keywords for this review.”

Line 78-79: do you mean oxidation of a methyl group or addition of a supplementary carboxyl moiety? Be more specific in your language usage. Hence correct to “catalyzes the oxidation of a methyl group”.

Line 79: these structures have not been disclosed in the prior list (lines 50-53) nor in the figure S1?

Line 80: “hydroxylation of the polyol region”? Suggested: “hydroxylation leading to the polyol region”.

Line 81: correct to “or the addition of epoxy groups as seen with AmphL, NysL, TetrK, and PimD”.

Line 81: in addition, these new derivatives are undisclosed till now (same remark as for line 79).

Legend figure 1:

Line 83: correct to “post-synthesis modifying proteins”.

Line 84: correct to “carboxylating CYP450 enzymes” (the CYP450 enzyme is not carboxylated itself).

Line 85: in analogy correct to “hydroxylating or epoxidizing CYP450 enzymes”.

Line 88: correct to “by one of the enzymes”.

Line 96: delete “some”.

Line 98: correct to “reduction or improvement in activity”.

Line 107: correct to “which were used”.

lines 111-116: all these details are superfluous for this review as we do not know at all what these various structures look like or what their modification entails.

line 117 describes these are decarboxylated derivatives, but why then 5 different BSG derivatives are mentioned and what are their structural differences?

Line 143: correct to “…of these polyene macrolides are inconsistent upon decarboxylation, …”.

Line 145: “showed upward trend”. In contrast with which is stated, their hemolytic activities were reduced as a higher concentration is needed for hemolytic activity! Hence, correct to “their toxicity decreased, as shown by reduced hemolytic activities (HCs)”.

Lines 166-168: it is claimed the carboxyl groups are important, but which interactions the carboxyl groups are involved in?  This is not clearly mentioned in the manuscript.

Furthermore, the interactions of the carboxyl groups with the sterols lead to “hydrogen bond networks involving the amino groups of mycosamine, water molecules, and hydroxyl groups on sterols”? There are no such groups on sterols. The whole paragraph needs rephrasing as the same problem is encountered in next lines.

Line 169: "ergosterol and cholesterol form a network with the same strength as that formed by antibiotics " what does this sentence actually mean? Literally you are implying that ergosterol and cholesterol are interacting with themselves?

Line 173: "acetate on the side chain carboxyl group"; there is no acetate "on" the carboxyl group. This should simply be "the carboxylate on C16 forms a salt bridge".

Line 175: “A water bridge hydrogen bond forms between the hydroxy energy on C13 and C2′ on mycosamine”.  The sentence is clearly nonsense, but from the figure 3, I gather what you mean. Hence, “A water bridged hydrogen bond forms between the epoxide oxygen on C13 and C2′-hydroxyl moiety on mycosamine”. In addition, in figure 3 also a H-H bond as drawn is nonsense. It is either the water oxygen atom that is bonding with the hydroxyl hydrogen, or vice versa. The figure needs to be corrected.

Lines 176-178: you infer at first there is an interaction with ergosterol, before the antibiotic gets its stabilized structure. What drives the interaction with ergosterol (cholesterol)? The narrative needs to be changed considerably.

Lines 182-185 should refer to figure 2 for structures.

line 185:  "showed the hight potential" and "in vivo experimental species "? I presume the authors meant to state "... showed their high potential and excellent effects on in vivo experiments in mice."

I stopped reviewing and correcting at this stage. All above remarks should indicate how the whole manuscript can and should be improved.

Reviewer 3 Report

The review article Antibiotics-2100935 ‘‘Effect of post-modification group on property and activity of 2 polyene macrolides’’ describe and compare the structure variations of different polyene macrolide antibiotics which could be useful in the development of antimicrobial/antifungal agents with high efficiency. 

Comments:

  • Table 1: Abbreviations may be defined in the table’s foot note.
  • Table 1: All Antifungal and Hemolytic Activities may be expressed in one format either in μmol/L or μg/mL as it will be easy for the reader to understand and compare the activities of various compounds.
  • Reference Gagos [6] may be added to figure 3.
  • The title of figure 4 may be repharased from ‘‘Several typical ideas of decarboxylation genetic engineering ’’to ‘‘Genetic engineering based strategies for decarboxylation of macrolide’’.
  • In the conclusion section, the author may also give statement on the public health importance of the review. How development of novel polyene macrolides could contribute in the control of microbial diseases burden and their treatment worldwide? )

Round 2

Reviewer 2 Report

The review of Liqin Qiao et al. has been substantially updated and improved, and language and phrasing at most parts now is fine. Only some small remarks are remaining.

The authors preferred to merge previous figures S1 and figure 2 into the new (and large) figure 1, which is fine. One of the structures however, still needs correction: herein, BSG013 should have the C16-COOH position instead of a methyl. BSG020 then is its non-oxidized analogue (having the C16-methyl moiety).

The sentence at line 209-210 sounds awkward and needs to be rephrased.

Line 334: correct to "fungi" (not fungis) or "fungal species".

Line 342: correct to “test” (single)

Line 356 and other places: systematically include a hyphen in the name between C2' and deoxy, to obtain "C2'-deoxyamphotericin B"

Line 466: correct to “accounts” (from accountes)
